# Experimental Study of a New Pneumatic Actuating System Using Exhaust Recycling

**Qihui Yu** [1,2,3,*], **Jianwei Zhai** [1], **Qiancheng Wang** [1], **Xuxiao Zhang** [1] **and Xin Tan** [1]

1   Department of Mechanical Engineering, Inner Mongolia University of Science and Technology, Baotou 014010, China; lyxpanbao@163.com (J.Z.); wqcmmt@163.com (Q.W.); zxx1120709066@163.com (X.Z.); tanxin@imust.edu.cn (X.T.)
2   State Key Laboratory of Fluid Power and Mechatronic Systems, Zhejiang University, Hangzhou 310027, China
3   Pneumatic and Thermodynamic Energy Storage and Supply Beijing Key Laboratory, Beijing 100191, China
*   Correspondence: yqhhxq@163.com

**Abstract:** Pneumatic actuating systems are an important power system in industrial applications. Due to exhaust loss, however, pneumatic actuating systems have suffered from a low utilization of compressed air. To recycle the exhaust energy, a novel pneumatic circuit was proposed to realize energy savings through recycling exhaust energy. The circuit consisted of three two-position three-way switch valves, which were used to control the exhaust flows into a gas tank or the ambient environment. This paper introduced the energy recovery configuration and working principles and built a mathematical model of its working process. Then, the mathematical model was verified by experiments. Finally, through experiments in which the air supply pressure, the critical pressure and the volume of the gas tank were regulated, the energy recovery characteristics of the pneumatic actuating system were obtained. Using the new circuit, the experimental results showed that the energy recovery efficiency exceeded 23%. When the air supply pressure was set to 5 bar, 6 bar, and 7 bar, the time required for pneumatic actuation to complete the three working cycles were 5.2 s, 5.3 s, and 5.9 s, respectively. When the critical pressure was set to 0 bar, 0.5 bar, 1 bar, and 1.5 bar, the times for pneumatic actuation to complete the three working cycles were 4.9 s, 5.1 s, 5.2 s, and 5.3 s, respectively. When the volume of the gas tank was set to 2 L, 3 L, 4 L, and 5 L, the number of working cycles was 3, 4, 5, and 6, respectively. This paper provides a new method of cylinder exhaust recycling and lays a good foundation for pneumatic energy savings.

**Keywords:** circuit; energy recovery; mathematical model; pneumatic actuating systems; working characteristics

## 1. Introduction

Compressed air is widely applied in many industries. According to some reports, compressed air systems account for approximately 10% of total industrial energy use [1]. In compressed air systems, pneumatic cylinders are one of the most important actuators and are widely applied in point-to-point transmission applications such as some automated production lines. Compared with other actuators such as electric cylinders, pneumatic cylinders have the advantages of low cost, ease of maintenance, mechanical simplicity, large output force/weight ratio, and environmental cleanliness. However, because of the compressibility of air, pneumatic cylinder drives exhibit highly nonlinear characteristics [2] and low energy efficiency [3]. In practical working systems, some studies have indicated that the energy efficiency of pneumatic cylinder systems can fall below 20% [4].

In recent years, some researchers have tried to save compressed air by using different methods. Based on a tracking control strategy, a pneumatic energy-saving circuit was presented by Wang and Gordon [5]. A pneumatic cylinder that used an open-loop control of a pneumatic position system using four on-off switching valves was presented by Krzysztof B.J. et al. [6]. The results indicated that a fast pneumatic positioning system with a low air

consumption and low price can be technically realized. Behrouz Najjari et al. [7] proposed fuzzy logic control strategies and verified the effectiveness of the control strategies by experiments. D. Saravanakumar et al., developed and implemented an interconnected system with two pneumatic cylinders for fast and accurate positioning [8]. Kaiji Sato et al., described a practical and intuitive controller design method for the precision positioning of pneumatic cylinder actuator stages. Three elements were added to the conventional continuous motion nominal characteristic trajectory following the controller. The position results generally indicate a positioning error of 50 nm, which was equal to the sensor resolution [9]. Li et al., used an inflatable accumulator to absorb the discharged compressed air, and then pressurized the air to a higher pressure through the pressurization valve for reuse, saving 40% of the compressed air [10]. Based on a discretization of the state equation of air, a motion equation, a continuity equation, and an energy equation, the pressure losses and time delay through a long connected pipeline were estimated. Experimental results showed that the distributed model could improve position accuracy [11]. Hongwang Du et al., used the bridge circuit to increase the energy efficiency of a single cylinder by 50% to 70%, they also improved the circuit by adding bypass valves and adopting better control algorithms. The results showed that the energy efficiency of a single cylinder can be further improved by modification 55~87% [12–14]. In order to simplify the implementation of energy-saving circuits for industrial end-users, Paul Harris et al., proposed a software program. A selection process of optimal design parameters and control input variables for the pneumatic cylinder were automated by software [15]. Teradihima et al., found that the effective utilization of exhaust air was most effective achieving energy savings for pneumatic equipment [16]. Yang et al., proposed a boost regulator, which can save 5% to 10% of energy [17]. Harris et al. [18] and Hepke et al. [19] reduced air consumption by adjusting the gauge pressure of the air supply in the system, that is, reduced the overpressure in the cylinder piston retraction phase. This can save nearly 30% of the input energy. Shi Yan et al. [20–22] proposed an expansion energy enhancer, which cuts off the air supply before the end of the piston stroke and uses air expansion to drive the piston to complete the remaining stroke. Similarly, many other methods are also used to improve the energy efficiency of cylinders, such as differential drive [23], recovering energy through a rubber bladder and storing strain energy [24], or reusing exhaust gas for power generation [25]. However, these methods make the circuit more complicated and cause speed fluctuations, which are harmful to the service life of the cylinder [26].

In most situations, exhaust air energy and expansion energy are not used in traditional pneumatic circuits; therefore, making full use of the exhaust energy and expansion energy have become key paths for enhancing the energy efficiency of pneumatic cylinders. Variable intake pressure control is a typical method used to enhance the effect of compressed air [27]. However, this method increases system costs and affects the output force. For some industrial end-users such as those who use clamped pneumatic cylinders, the output force during the power stroke is the important factor, and accurate piston control of the pneumatic cylinder return trip is not required. Making full use of exhaust energy is an effective approach.

The purpose of this research is to provide an efficient pneumatic actuator system by recycling exhaust energy and study its working characteristics. In our previous research, a sensitivity analysis of parameters was conducted using a dimensionless method [28]. In this paper, a new pneumatic circuit for cylinder motion system, an exhaust energy recovery (EER) pneumatic cylinder, is proposed that makes use of the exhaust power during the exhaust stroke to enhance energy efficiency. The new pneumatic system consists of three control valves that are used to carry out exhaust recovery and a gas tank that collects exhaust gas. First, the loop and working principle of the EER pneumatic system is introduced. Second, the basic mathematical model of the system is established. Then, a test platform for the novel circuit is built and investigated to verify the mathematical model. Finally, through experiments, when the air supply pressure, critical pressure, and volume of the gas tank are regulated within the ranges of 5–7 bar, 0.5–1.5 bar, and 2–5 L,

respectively. The kinematic characteristics of the piston and energy savings in the gas tank are studied. The research lays a good technical foundation for further research on highly efficient pneumatic actuating system.

## 2. Configuration and Working Principles of EER Pneumatic Circuits

The configuration of the EER system is composed of a cylinder, three solenoid valves, two pressure sensors, one gas tank, one check valve and a controller which are shown in Figure 1. The air supply pressure of the cylinder is controlled by a pressure regulator. The valves are named $Val_1$, $Val_2$, and $Val_3$, which represent solenoid valves 4, 5, and 11, respectively. $Val_1$ and $Val_2$ are connected to chamber a, and $Val_3$ is connected to chamber b. The air pressure values in chambers b and the gas tank, which are monitored by pressure sensors 1 and 2, are defined as $p_{cb}$ and $p_t$, respectively. The speed of the pneumatic cylinder is regulated by a one-way restrictive valve.

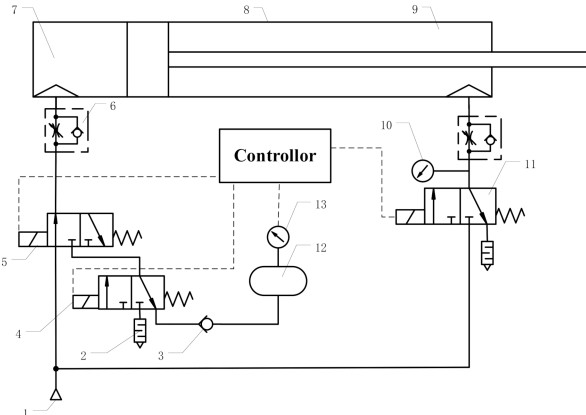

**Figure 1.** Structure of the exhaust energy recovery (EER) system. (1). Air source; (2). Silencer; (3). Check valve; (4). Solenoid valve 1; (5). Solenoid valve 2; (6). One-way restrictive valve; (7). Left pneumatic cylinder; (8). Pneumatic cylinder; (9). Right pneumatic chamber; (10). Pressure sensor 1; (11). Solenoid valve 3; (12). Gas tank; (13). Pressure sensor 2.

When the piston reaches its left travel destination, the compressed air enters into the left pneumatic chamber through $Val_2$ and pushes the piston to the right. At this time, the air in the right cylinder chamber flows to the environment. When the piston reaches its right extreme position, the solenoid valve $Val_3$ changes its state, and the air flows into the right cylinder chamber through $Val_3$ and pushes the piston to the left. At this time, if the pressure difference between the air pressure in the right pneumatic chamber $p_{cb}$ and the air pressure in the gas tank $p_t$ is larger than a critical pressure $p_b$, and the air in the left cylinder chamber flows into the gas tank; if the pressure difference is smaller than the critical pressure $p_b$, the air in the left cylinder chamber enters the ambient environment.

## 3. Mathematical Models of EER System

To facilitate this research, the following assumptions are made:

(1)　The working fluid (air) of the system follows all ideal gas laws.
(2)　There is no leakage between the chambers, and the effective areas of all intake and exhaust ports are the same.
(3)　The supply temperature is equal to the atmospheric temperature.
(4)　The air flowing into and out of chambers a and b is a stable one-dimensional flow that is equivalent to the flow of air through the nozzle contraction.
(5)　The working process is an isothermal process.

Using the above assumption, a mathematical model of the EER system is obtained according to the continuity equation, energy equation, state equation, and dynamic equation.

### 3.1. Continuity Equation of the EER System

The left and right pneumatic cylinder chambers are defined as chambers a and b, and the air masses in chambers a and b are represented here by $m_a$ and $m_b$, respectively. The air mass in the gas tank is represented by $m_t$. The mass flow through the solenoid valve $Val_i$ is represented by $m_i$ (i = 1,2,3). Based on the ratio $p_l/p_h$ and the critical pressure ratio, $b$, the air mass flow equation for the flow through a restriction can be written as follows [29]:

$$q = \begin{cases} \frac{A_e p_h B}{\sqrt{T_h}} \left[ \left(\frac{p_l}{p_h}\right)^{\frac{2}{\kappa}} - \left(\frac{p_l}{p_h}\right)^{\frac{\kappa+1}{\kappa}} \right] & \frac{p_l}{p_h} > b \\ \frac{A_e p_h D}{\sqrt{T_h}} & \frac{p_l}{p_h} \le b \end{cases} \tag{1}$$

where

$$B = \sqrt{\frac{2\kappa}{R(\kappa - 1)}} \tag{2}$$

$$D = \left(\frac{2}{\kappa + 1}\right)^{\frac{1}{\kappa-1}} \sqrt{\frac{2\kappa}{R(\kappa + 1)}} \tag{3}$$

where $A_e$ is the effective area (m²); $p_h$ is the pressure of the upstream side (Pa); $p_l$ is the pressure of the downstream side (Pa); $\kappa$ is the specific heat ratio (null); $T_h$ is the temperature of the upstream side (K); and $R$ is the gas constant, 287 J/(kg·K).

### 3.2. Energy Equation of the EER System

Because leakage is ignored in the working process, the driving chambers do not charge and discharge air simultaneously. Consequently, the energy equation of the left pneumatic chamber can be described as follows:

For the charge process:

$$C_v m_a \frac{dT_a}{dt} = S_a \cdot h_a(T_{amb} - T_a) + (C_p T_{amb} - C_v T_a)q_a - p_a A_a u \tag{4}$$

For the discharge process:

$$C_v m_a \frac{dT_a}{dt} = S_a \cdot h_a(T_{amb} - T_a) + R q_a T_a - p_a A_a u \tag{5}$$

where $t$ is time (s); $S_a$ is the heat transfer area of the chamber a (m²); $C_v$ is the heat capacity at a constant volume (J/(kg·K)); $T_a$ is the air temperature in chamber a; $T_{amb}$ is the atmospheric temperature (K); $h_a$ is the heat transfer coefficient for the left side (W/(m²·K)); $q_a$ is the air mass flow in chamber a (g/s); $p_a$ is the pressure in chamber a (Pa); $A_a$ is the area of the piston in chamber a (m²); $u$ is the velocity of the piston (m/s); $m_a$ is the air mass of chamber a (g).

The energy equation for the right pneumatic chamber can be given by the following equations:

For charge process:

$$C_v m_b \frac{dT_b}{dt} = S_b \cdot h_b(T_{amb} - T_b) + (C_p T_{amb} - C_v T_b)q_b - p_b A_b u \tag{6}$$

For discharge process:

$$C_v m_b \frac{dT_b}{dt} = S_b \cdot h_b(T_{amb} - T_b) + R q_b T_b - p_b A_b u \tag{7}$$

where $S_b$ is the heat transfer area of chamber b (m²); $T_b$ is the air temperature in chamber b (K); $h_b$ is the heat transfer coefficient for the left side (W/(m²·K)); $q_b$ is the air mass flow in

chamber b (g/s); $p_b$ is the pressure in chamber b (Pa); $A_b$ is the area of the no rod end (m²); and $m_b$ is the air mass of chamber b (g).

### 3.3. State Equation of the EER System

According to the state equation of ideal gases, air pressure in each chamber can be expressed:

$$\frac{dp}{dt} = \frac{1}{V}\left[\frac{pV}{T} \cdot \frac{dT}{dt} + RTq - pAu\right] \tag{8}$$

where, $V$ is the volume (m³); $p$ is the pressure of the compressed air (Pa); $T$ is the temperature of the compressed air (K); and $A$ is the area of the piston in the pneumatic chamber (m²).

### 3.4. Dynamic Equation

The velocity of the piston is derived from Newton's second law of motion. In this paper, the friction force model is considered to be the sum of the Coulomb friction and viscous friction [30]. The viscous friction force is considered to be a linear function of piston velocity. The coordinate system for an asymmetric pneumatic cylinder is illustrated in Figure 2.

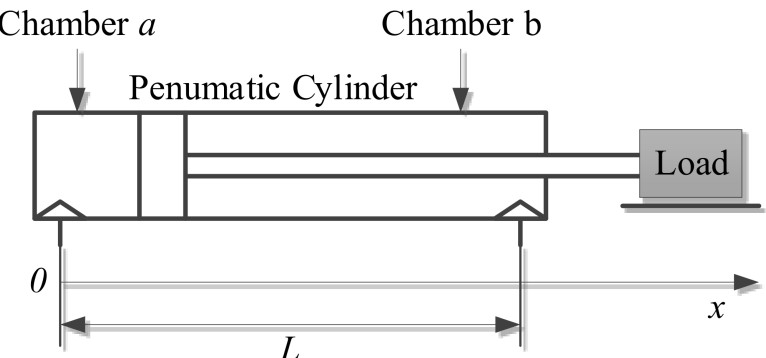

**Figure 2.** The coordinate system of a pneumatic cylinder.

The right side is considered to the positive direction of the vector. The motion equation of the piston can be given by the following equation:

$$\frac{d^2x}{dt^2} = \begin{cases} \frac{1}{M}\left(p_{ca}A_{ka} - p_{cb}A_{kb} - p_aA_r - F_f\right) & x \neq 0, L \\ 0 & x = 0, L \end{cases} \tag{9}$$

where $M$ is the mass of load (kg); $A_r$ is the effective area of the piston rod (m²); $p_a$ is the external environment pressure (Pa); and $A_{ki}$ is the effective area of the piston (m²).

$F_f$ is the friction of the cylinder system. In Du et al. [12], the friction was calculated by

$$F_f = \begin{cases} \beta \cdot u + \left[f_c + (f_s - f_c)e^{[-(u/u_s)^\delta]}\right]\text{sgn}(u) & u \leq u_e \\ \mu \cdot u & u > u_e \end{cases} \tag{10}$$

where $\beta$ is the viscous coefficient of the pneumatic cylinder (N/(m/s)); $f_c$ is the Coulomb friction force (N); $f_s$ is the maximum static friction force (N); $u$ is the velocity of the piston (m/s); $\mu$ is the dynamic friction factor; $u_s$ is the Stribeck velocity (0.0001–0.1) (m/s); $\delta$ is an arbitrary index (0.5–2); and $u_e$ is the critical velocity (m/s).

### 3.5. Gas State in Air Recovery Tank

The variations in air temperature and air pressure in the gas tank can be obtained by [30]:

$$\begin{cases} \frac{dp_t}{dt} = \frac{R}{C_v V_t} \left[ q_t C_p T_a + h_t S_t (T_{amb} - T_t) \right] \\ \frac{dT_t}{dt} = \frac{RT_t}{C_v p_t V_t} \left[ q_t C_p T_a - q C_v T_t + h_t S_t (T_{amb} - T_t) \right] \end{cases} \tag{11}$$

where $V_t$ is the volume of the air recovery tank (m³); $T_t$ is the air temperature in the gas tank (K); $h_t$ is the heat transfer coefficient of the gas tank (W/(m²·K)); $q_t$ is the air mass that flows into the gas tank (g/s); $S_t$ is the heat transfer area of the gas tank (m²); and $C_p$ is the heat capacity at constant pressure (J/(kg·K)).

### 3.6. Recovered Energy and Efficiency

According to [21], the available energy of flowing compressed air presents energy that can be theoretically converted to mechanical work at atmospheric pressure. The recovery available energy in the gas tank can be calculated by.

$$E = p_{amb} V_{amb} \ln \frac{p_t}{p_{amb}} \tag{12}$$

where $p_{amb}$ is the ambient pressure (Pa) and $V_{amb}$ is the volume of air at the standard state (m³).

Recovered efficiency can be expressed as:

$$\eta = \frac{E}{E_W} \times 100\% \tag{13}$$

where $E_W$ is the consumption energy during the working cycle (J), which can be calculated by.

$$E_W = p \ln \frac{p}{p_a} \int q dt \tag{14}$$

### 3.7. Algorithm

The basic parameters are shown in Table 1, and the above mathematical model can be built using the MATLAB/Simulink (R2016a version). The Simulink diagram is shown in Figure 3.

**Table 1.** Basic parameters of the EER system.

| Symbol | Quantity | Value |
|--------|----------|-------|
| R | Gas constant | 287 J/kg·K |
| $T_{amb}$ | Ambient temperature | 293 K |
| $p_{amb}$ | Ambient pressure | 101,000 Pa |
| $h_a$ | Heat transfer coefficient of the left side | 30 W/(m²·K) |
| $h_b$ | Heat transfer coefficient of the right side | 20 W/(m²·K) |
| $h_t$ | Heat transfer coefficient of the gas tank | 20 W/(m²·K) |
| $A_{ka}$ | Effective area of piston of chamber a | 0.0031 m² |
| $A_{kb}$ | Effective area of piston of chamber b | 0.0028 m² |
| $\kappa$ | Specific heat ratio | 1.4 |
| M | Mass of load | 30 kg |
| $F_s$ | Maximum static friction force | 110 N |
| $F_c$ | Coulomb friction force | 80 N |
| $\beta$ | Viscous coefficient of pneumatic cylinder | 10 N/(m/s) |
| $\mu$ | Dynamic friction factor | 5 |
| L | Pneumatic cylinder stroke | 0.2 m |
| $A_e$ | Effective area | 0.0001 m² |
| $V_t$ | Volume of air recovery tank | 0.004 m³ |

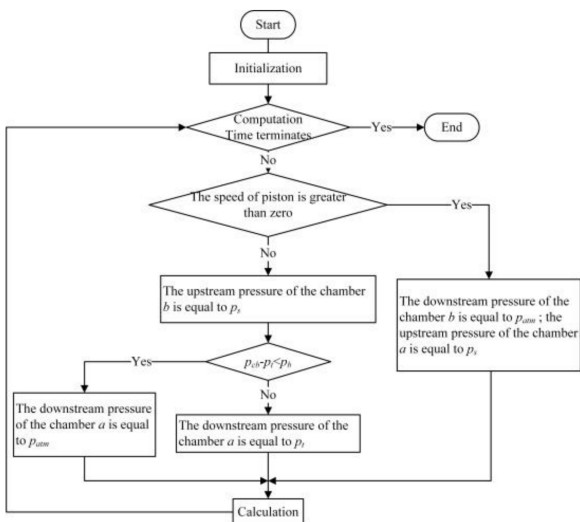

**Figure 3.** Algorithm for the mathematical model ($p_{cb}$ is the air pressure in chamber b; $p_{atm}$ is the environmental pressure; $p_t$ is the air pressure in the recovery tank; $p_b$ is the difference between $p_s$ and $p_t$; $p_s$ is the air supply pressure.).

## 4. Experimental Verification of the Mathematical Model

To verify the accuracy of the mathematical model, an experimental platform is built. The details are shown in Figure 4. The compressed air is controlled by a regulator (AR5000-10, KEH, Shanghai, China), a filter (AF5000-10, KEH, Shanghai, China), and a lubricator (AL5000-10, KEH, Shanghai, China). Its main function is to reduce the pressure to a fixed value. Three pressure sensors (ISE40A, SMC, Tokyo, Japan) are used to monitor the air pressure in chamber a, chamber b, and the gas tank. A displacement sensor (KA-300, SINO, Guangzhou, China) is used to monitor the piston displacement. A force sensor (ZNLBU-11, CHINO SENSOR, Bengbu, China) is used to measure the output force of the pneumatic cylinder. A NI data acquisition card (USB6211, NI, Austin, Texas, USA) and a computer (X250, Lenovo, Beijing, China) are used to obtain experimental data.

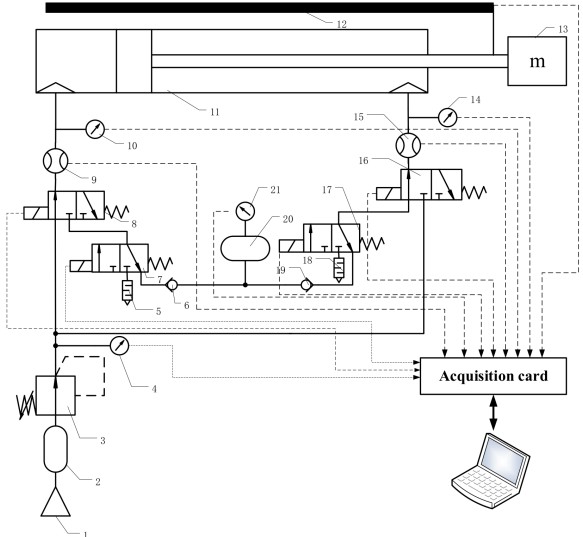

**Figure 4.** Schematic diagram of the experimental system:(1) air source; (2), (20) gas tank; (3) reduction valve; (4), (10), (14), (21) pressure sensor; (5), (18) muffler; (6),(19) check valve; (7), (8), (16),(17) two position three-way solenoid valve; (9), (15) flow sensor; (11) cylinder; (12) displacement sensor; (13) load.

The test range of pressure sensor and force sensor are 0–1 MPa, 3–3000 N, respectively. And the accuracy of pressure, displacement and force is 1%, ±5 μm and ±0.05% F.S., respectively. So the maximum absolute error of pressure, displacement and force are 0.01 MPa, ±5 μm and ±1.5 N, respectively.

A platform for the EER system, which is shown in Figure 5, was built and designed to measure the output force, piston displacement. The compressed air source was opened, then the air pressure was set to a fixed value by regulator. The EER system controlled and saved data with LabVIEW (2018 professional version). Three solenoid valves were controlled to realize reciprocating movement of the cylinder. The input air pressure ($p_s$), the critical pressure ($p_b$) and the volume of gas tank ($V_t$) were adjusted during the experimental process.

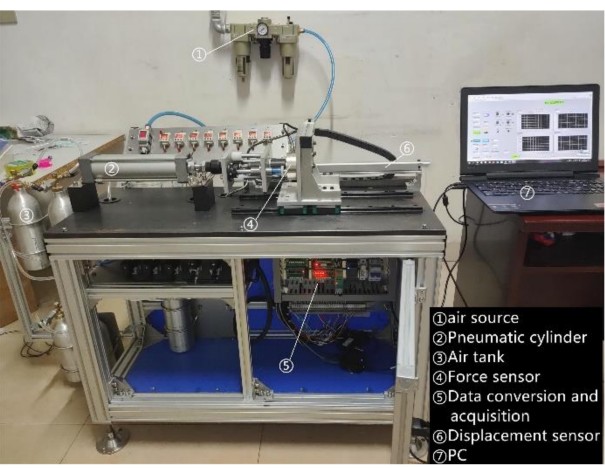

**Figure 5.** Test bench.

In the experimental study, other parameters are held constant, the air supply pressure $p_s$ is set at 5 bar, 6 bar, and 7 bar, the critical value of pressure $p_b$ is set at 0.5 bar, 1 bar, and 1.5 bar, the volume of gas tank $V_t$ is set at 2 L, 3 L, 4 L, and 5 L. Experiments are performed under each condition. Therefore, thirty-six groups of experiments are carried out totally.

When the gas supply pressure is 6bar, the critical pressure is 1.5 bar, and the gas tank is 4 L, the air pressure in the gas tank is shown in Figure 6. It is obvious that with the number of cycle index increasing, the air pressure in gas tank gradually increases. During the 5 cycles, the air pressure in gas tank reaches 4.36 bar.

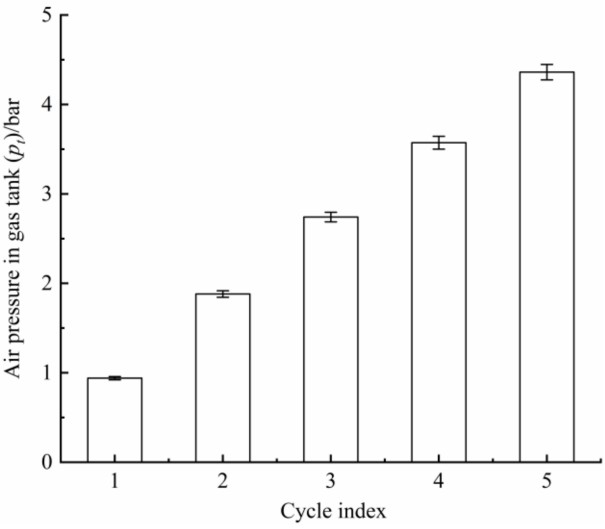

**Figure 6.** The relationship between cycle index and air pressure in gas tank.

When the supply pressures are set to approximately 5 bar, 6 bar, and 7 bar, the volume of the gas tank, $V_t$ and the critical value of pressure $p_b$ are set to 4 L and 1 bar, respectively. Three representative experiments and their simulation results are shown in Figure 7. Figure 7 shows that the pressure of the gas tank $p_t$ increases with the number of cycles. The experimental pressure is basically consistent with the simulation results under the same air supply pressure, but there are slight differences: the experimental pressure is slightly higher. The causes of the differences are analyzed as follows. During the experimental process, there is obvious pressure fluctuation in the air supply pressure, which is shown in Figure 8, and the amplitude of the fluctuation depends on the working condition. Additionally, the solenoid valve has a delayed effect, and the size of the valve port is not fixed.

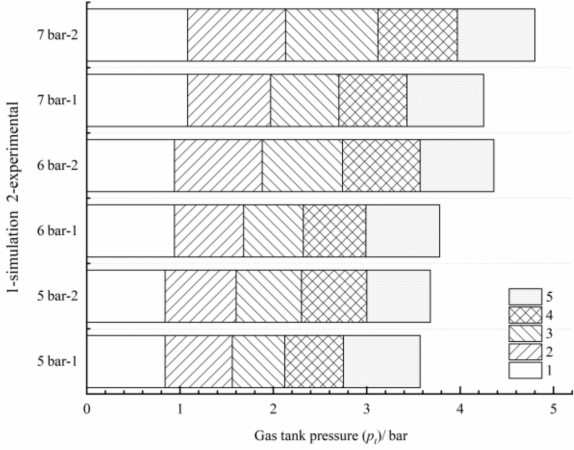

**Figure 7.** Comparison between experiment and simulation (graphic legend 1, 2, 3, 4, 5 represents cycle index).

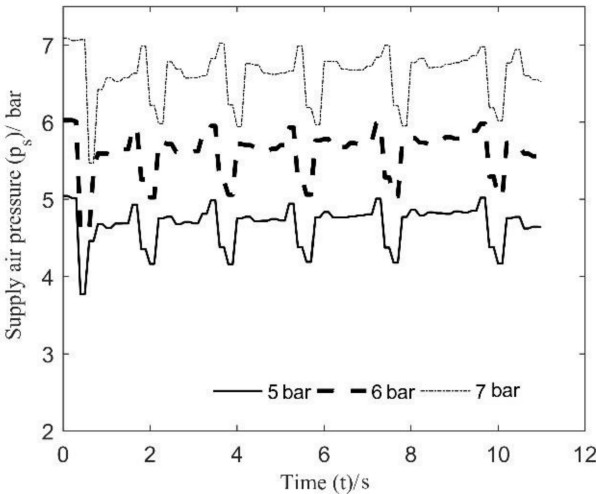

**Figure 8.** Air Supply pressure curves.

As shown in Figure 9, it is gas tank pressure and piston displacement results of the experiments when the supply pressure, critical pressure, and volume of the gas tank are set to 6 bar, 1 bar, and 4 L, respectively. It is obvious that the gas tank pressure increases with increasing cycle times. Meanwhile, as the gas tank pressure increases, the pressure difference between the gas tank and the cylinder exhaust cavity is reduced, and the time required to complete a cycle increases with the cycle time increases.

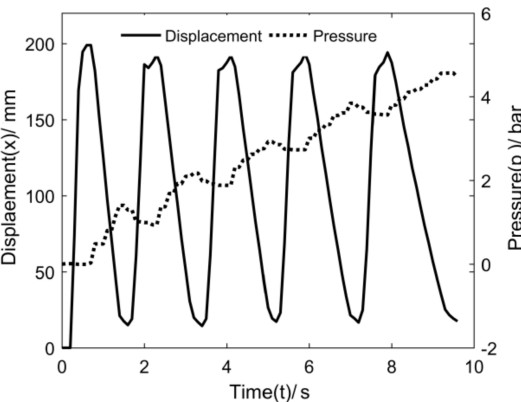

**Figure 9.** Gas tank pressure and piston displacement curves.

## 5. Study of the EER System Characteristics

Based on the authors' previous study [28], the energy savings of the EER system are mainly determined by the air supply pressure ($p_s$), the difference between the supply pressure and the gas tank pressure ($p_b$), and the volume of the gas tank, ($V_t$). To study the effect of the three parameters on the consumption time for the exhaust stroke, piston displacement was investigated under different $p_s$, $p_b$, and $V_t$ conditions by experiment. Meanwhile, the energy savings of the EER system were obtained by calculation based on the experimental results. The ranges of $p_s$, $p_b$, and $V_t$ are set to 5–7 bar, 0.5–1.5 bar, and 2–5 L, respectively.

During the experiment, each parameter is changed while the other parameters are held constant. The section can serve as the basis for further multi-objective optimization to determine the trade-off between the energy savings and consumption time.

### 5.1. Influence of the Air Supply Pressure

In many industries, the air supply pressure is usually regulated to meet the load requirement. The characteristics of the EER system are investigated when the critical pressure ($p_b$) is set to 1bar, the volume of gas tank ($V_t$) is set to 4 L, and the air supply pressure ($p_s$) is set to 5 bar, 6 bar, and 7 bar.

Figure 10 shows the piston displacement curves, and we can see the variation trend in the curves. At different air supply pressures, the time required for pneumatic actuation to complete the three working cycles is different. When the air supply pressure is set to 5 bar, 6 bar, and 7 bar, the times are equal to 5.2 s, 5.3 s, and 5.9 s, respectively. The primary cause of the differences is that when the air supply pressure increases, the pressure of the exhaust chamber will increase during the air recovery process, which can reduce the speed of the piston.

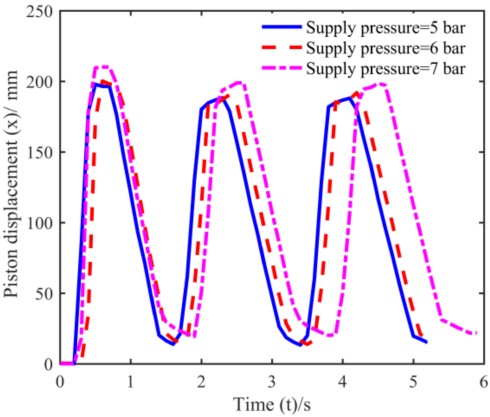

**Figure 10.** Piston displacement curves under different air supply pressures.

As is shown in Figure 10, if we increase air supply pressure, the time required for pneumatic actuation to complete working cycle will increase apparently.

As shown in Figure 11a, as the air supply pressure increases from 5 bar to 7 bar, the recovery energy increases. When the critical pressure increases from 0.5 bar to 1.5 bar, the recovery energy decreases because the greater the critical pressure is, the smaller the pressure difference between the intake chamber and the exhaust chamber, which leads to a decrease in the mass flow of gas entering the gas tank and a decrease in the pressure change of the gas, thus showing a downward trend. However, when the critical pressure increases from 1 bar to 1.5 bar, the recovery energy slightly decreases when the air supply pressure is set to 7 bar.

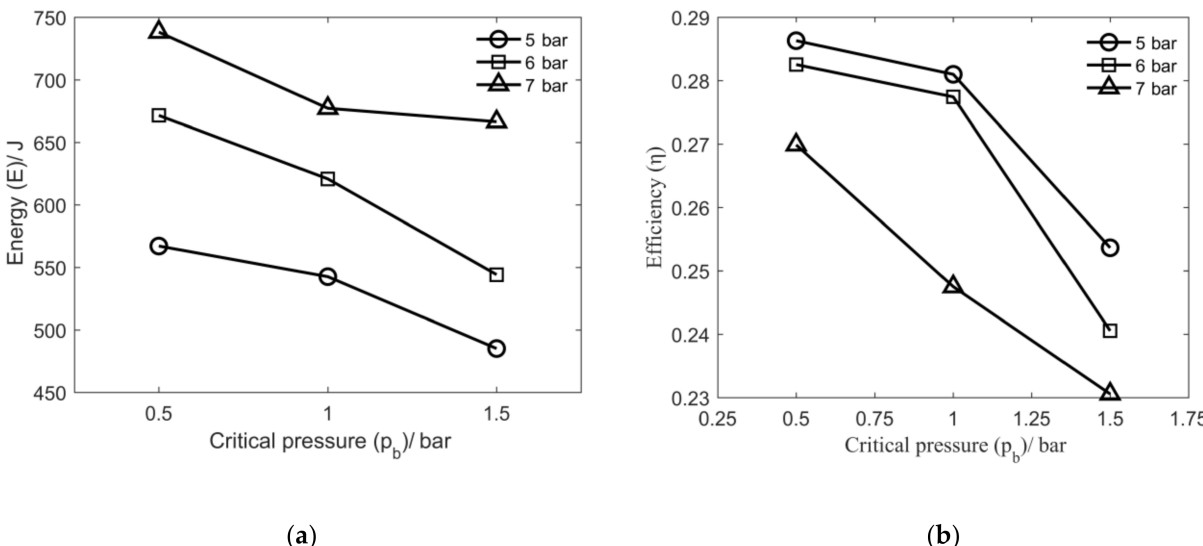

(**a**)                                                   (**b**)

**Figure 11.** Relationship between the air supply pressure, critical pressure, recovery energy and efficiency. (**a**) Energy vs. critical pressure, (**b**) Efficiency vs. critical pressure.

As shown in Figure 11b, as the air supply pressure increases from 5 bar to 7 bar, the recovery energy efficiency decreases. When the critical pressure increases from 0.5 bar to 1.5 bar, the recovery energy efficiency decreases. It is clear that the energy recovery efficiency is about 23–28.7% by using the EER system. The results accord with literature research. For example, Yang et al., designed a control scheme which can save 12–28% energy [31] and compressed air energy saving is about 10.9–29.5% combined with the digital sliding mode [32].

From Figure 11, we can get that the air supply pressure will influence the saving energy and efficiency. Lower air supply pressure will decrease the saving energy, but energy recovery efficiency will increase.

### 5.2. Influence of the Critical Pressure

The critical pressure directly reflects the principle and the extent of the pressure in the gas tank. The characteristics of the EER system are investigated when the air supply pressure ($p_s$) is set to 6 bar, the volume of the gas tank ($V_t$) is set to 4 L, and the critical pressure ($p_b$) is set to 0 bar, 0.5 bar, 1 bar, and 1.5 bar.

Figure 12 shows the piston displacement curves, and we can see the variation trend in the curves. At different critical pressures, the time required for pneumatic actuation to complete the three working cycles is different. When the critical pressure is set to 0 bar, 0.5 bar, 1 bar, and 1.5 bar, the times are equal to 4.9 s, 5.1 s, 5.2 s, and 5.3 s, respectively. The primary cause of the differences is summarized as follows. When the critical pressure is set to 0 bar, the air in the exhaust chamber directly flows into the environment, so the velocity of the piston is the fastest. When the critical pressure is set higher, the speed of

the piston decreases. When the critical pressure is set to 0.5 bar and 1 bar, the air in the exhaust chamber flows into the environment owing to the fluctuation of the air supply pressure which is shown in Figure 8. When the critical pressure is set to 1.5 bar, the air in the exhaust chamber flows into the gas tank. Under this condition, the pressure of the exhaust chamber will increase.

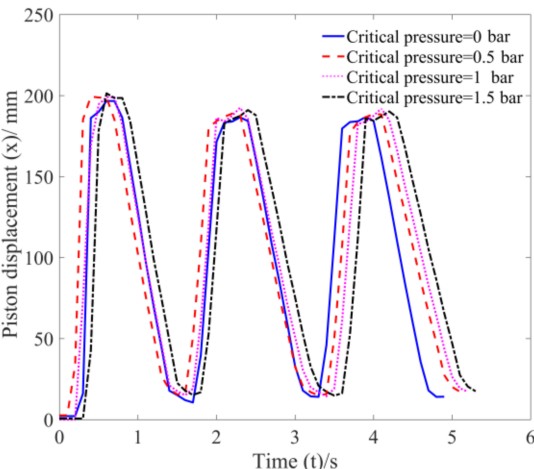

**Figure 12.** Piston displacement curves under different critical pressure.

As is shown in Figure 12, if we enlarge the critical pressure, the time required for pneumatic actuation to complete the working cycle will increase. So, for the practical application of the EER system, we can regulate the critical pressure to meet the working cycle time.

### 5.3. Influence of the Gas Tank Volume

The gas tank directly reflects the pressure of the exhaust chamber. The characteristics of the EER system are studied when the air supply pressure ($p_s$) is set to 6 bar, the critical pressure ($p_b$) is set to 1 bar, the volume of gas tank ($V_t$) is set to 2 L, 3 L, 4 L, and 5 L.

Figure 13 shows the piston displacement curves, and we can see the variation trend in the curves. The time it takes the pressure of the gas tank ($p_t$) to reach the setting pressure is measured at different gas tank volumes. When the volume of the gas tank is set to 2 L, 3 L, 4 L, and 5 L, the number of working cycles is 3, 4, 5, and 6, respectively. The primary cause of the differences is that the larger the volume of the tank is, the more the pressure-building time increases.

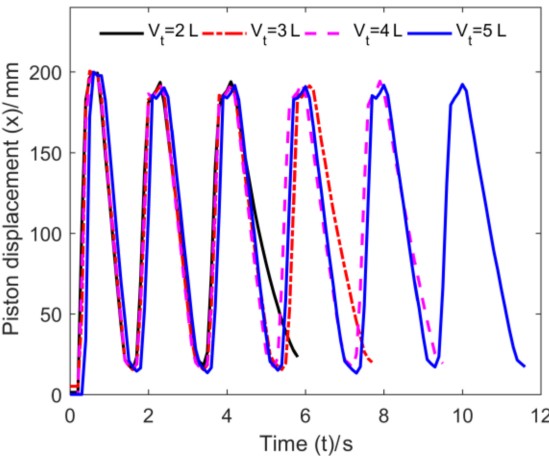

**Figure 13.** Piston displacement curves under different tank volumes.

From Figure 13, an increase in the volume of the gas tank will increase the pressure-building time. While we cannot set the volume of the gas tank too small to obtain sufficient energy. However, when the volume of the gas tank is set larger, the space to the EER will increase.

## 6. Conclusions

To improve the efficiency of traditional pneumatic actuating systems, a new kind of pneumatic circuit, an EER pneumatic circuit, is proposed in which energy is recycled by using a gas tank. The conclusions of this study are listed as follows:

(1) An EER pneumatic circuit is proposed to recycle the exhaust energy.
(2) The simulation results are in good agreement with the experimental results, which proves that the mathematical model is effective and accurate.
(3) The EER characteristics are influenced by the air supply pressure, critical pressure, and volume of the gas tank. In each case, the energy recovery efficiency exceeds 23%.
(4) At different critical pressures, the time required for pneumatic actuation to complete the three working cycles is different. When the critical pressure is set to 0 bar, 0.5 bar, 1 bar, and 1.5 bar, the times are equal to 4.9 s, 5.1 s, 5.2 s, and 5.3 s, respectively.
(5) At different air supply pressure, the time required for pneumatic actuation to complete the three working cycles is different. When the air supply pressure is set to 5 bar, 6 bar, and 7 bar, the time is equal to 5.2 s, 5.3 s, and 5.9 s, respectively.
(6) When the volume of the gas tank is set to 2 L, 3 L, 4 L, and 5 L, the number of working cycles is 3, 4, 5, and 6, respectively.

Overall, the proposed EER system can achieve energy savings. During the actual process, the energy recovery efficiency and working cycle time should be considered, so the parameters should be optimized based on requirements. Therefore, this study provides a good foundation for future research on pneumatic energy-saving systems.

**Author Contributions:** Conceptualization, Q.Y. and J.Z.; methodology, Q.Y.; software, Q.Y.; valida-tion, Q.Y., J.Z. and Q.W.; formal analysis, Q.Y.; investigation, J.Z.; resources, Q.W.; data curation, Q.Y.; writing—original draft preparation, X.Z.; writing—review and editing, J.Z.; visualization, X.T.; supervision, X.Z.; project administration, Q.W.; funding acquisition, Q.Y. All authors have read and agreed to the published version of the manuscript.

**Funding:** This research was funded by the National Natural Science Foundation of China (Grant 52065054) and Open Foundation of the State Key Laboratory of Fluid Power and Mechatronic Systems (Grant GZKF-201804). It was also funded by Outstanding Young Scientists in Beijing (Grant BJJWZYJH01201910006021) and the Natural Science Foundation of Inner Mongolia (Grant 2018BS05003).

**Data Availability Statement:** The data that support the findings of this study are available from the corresponding author upon reasonable request.

**Conflicts of Interest:** The authors declare no conflict of interest.

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
