# Peer review of "Experimental Study of a New Pneumatic Actuating System Using Exhaust Recycling"

_sustainability, doi:10.3390/su13041645_

Round 1
Reviewer 1 Report
Please see the attachment.

Author Response
Dear reviews,
Thanks for your useful comments and so kind suggestions on the detail of our manuscript. We have modified the manuscript accordingly, and the detailed corrections are listed below:
Reviewer#1, Concern # 1: The manuscript requires deep literature review in order to discuss the present and past results
Author response: The authors agreed and adopted the reviewers' suggestions. We reviewed more literature including 8 recommended papers, summarized and discussed the current findings, and finally revised the manuscript.
Author action: We read more literature, which is explained in detail in the introduction of the manuscript (lines 63-82).
Best wishes,
Yours sincerely,
Reviewer 2 Report
Section 3. Mathematical Model
The authors must justify their assumption of the heat transfer coefficients (ha, hb, and ht). The assumption that the heat transfer coefficients are constant is fine. However, heat transfer coefficients are obtained from experimental correlations (likely in the form of a Nusselt) number. The experimental correlations used to obtain the heat transfer coefficients should be shown.
Section 4. Experimental Verification
The accuracy of the equipment is addressed (lines 228 - 229). However, the propagation of uncertainty must still be addressed. The individual experimental uncertainties of the measurement equipment propagate to uncertainties in the final result. The authors should also add error bars to figures as appropriate.
The authors should perform a statistical analysis to quantify the precision of their results.
Author Response
Dear reviews,
Thanks for your useful comments and so kind suggestions on the detail of our manuscript. We have modified the manuscript accordingly, and the detailed corrections are listed below:
Reviewer#2, Concern # 1: The authors must justify their assumption of the heat transfer coefficients (ha, hb, and ht). The assumption that the heat transfer coefficients are constant is fine. However, heat transfer coefficients are obtained from experimental correlations (likely in the form of a Nusselt) number. The experimental correlations used to obtain the heat transfer coefficients should be shown
Author response: It should be noted that in the third part of the manuscript, five hypotheses are made for the simulation research in the first place (line 139).The fifth hypothesis is "The working process is an isothermal process". When the working process is an isothermal process, the influence of heat transfer coefficient on the system is zero. So in this manuscript alone, the calculation of heat transfer coefficient be considered as a superfluous process.
Author action: No change.
Reviewer#2, Concern # 2: The accuracy of the equipment is addressed (lines 228 - 229). However, the propagation of uncertainty must still be addressed. The individual experimental uncertainties of the measurement equipment propagate to uncertainties in the final result. The authors should also add error bars to figures as appropriate
Author response: The authors agreed and adopted the reviewers' suggestions. The propagation of uncertainty have been discussed and the error bar has been given in Figure 6 in revised manuscript.
Author action: We discussed the propagation of uncertainty (lines 251-254), and error bar is shown in Figure 6 (lines 269-272, lines 284-285).
Best wishes,
Yours sincerely,
<Qihui Yu> et al.

Reviewer 3 Report
Dear authors,
Your paper investigates the pneumatic circuit to increase the exhaust efficiency. You provided the full envelope of modellings and equations. Those simulation modellings agree with the experimental results.
In the introduction, you summarized the related research/papers. Might better make tables to summarize those previous findings.
In conclusion, you might better write the summary of the research/paper instead of listing your findings.
I enjoyed this paper very much. I hope this paper will be accepted in the future."
Regards,
Author Response
Dear reviews,
Thanks for your useful comments and so kind suggestions on the detail of our manuscript. We have modified the manuscript accordingly, and the detailed corrections are listed below:
Reviewer#3, Concern # 1: In conclusion, you might better write the summary of the research/paper instead of listing your findings
Author response: We accepted and adopted the suggestion.
Author action: The introduction has been appropriately revised to highlight the conclusions of the reference literature.
Best wishes,
Yours sincerely,
<Qihui Yu> et al.
Round 2
Reviewer 1 Report
Literature cited is still not adequate to the present state of the art and contemporary results. There is no actual investigations published, where authors should compare such approaches. Therefore, it is impossible to evaluate the novelty of the work. I suggest to reject the paper in the present form.
Author Response
Dear reviews,
Thanks for your useful comments and so kind suggestions on the detail of our manuscript. We have modified the manuscript accordingly, and the detailed corrections are listed below point by point,and highlight the changes to the revised manuscript within the marked up manuscript by using yellow background:
Reviewer#1, Concern # 1: Literature cited is still not adequate to the present state of the art and contemporary results. There are no actual investigations published, where authors should compare such approaches. Therefore, it is impossible to evaluate the novelty of the work.
Author response: The authors agreed and adopted the reviewers' suggestions. We reviewed more literature and discussed the current findings, and finally revised the manuscript.
Author action: We read more literature and comparison between results with literature data have been done in revised manuscript.
Best wishes,
Yours sincerely,
<Qihui Yu> et al.
Reviewer 2 Report
Having addressed the reviewers' comments the manuscript is now acceptable for publication.
Author Response
Dear reviews,
Thanks for your useful comments and so kind suggestions on the detail of our manuscript.
Best wishes,
Yours sincerely,
<Qihui Yu> et al.